# INT reduction is a valid proxy for eukaryotic plankton respiration despite the inherent toxicity of INT and differences in cell wall structure

E. Elena García-Martín ¤*, Isabel Seguro, Carol Robinson

Centre for Ocean and Atmospheric Sciences, School of Environmental Sciences, University of East Anglia, Norwich, Norfolk, United Kingdom

¤ Current address: Ocean Biogeochemistry and Ecosystems, National Oceanography Centre, Southampton, Hampshire, United Kingdom
* elencia@noc.ac.uk

## Abstract

The reduction of 2-para (iodophenyl)-3(nitrophenyl)-5(phenyl) tetrazolium chloride (INT) is increasingly being used as an indirect method to measure plankton respiration. Its greater sensitivity and shorter incubation time compared to the standard method of measuring the decrease in dissolved oxygen concentration, allows the determination of total and size-fractionated plankton respiration with higher precision and temporal resolution. However, there are still concerns as to the method's applicability due to the toxicity of INT and the potential differential effect of plankton cell wall composition on the diffusion of INT into the cell, and therefore on the rate of INT reduction. Working with cultures of 5 marine plankton (*Thalassiosira pseudonana* CCMP1080/5, *Emiliania huxleyi* RCC1217, *Pleurochrysis carterae* PLY-406, *Scrippsiella* sp. RCC1720 and *Oxyrrhis marina* CCMP1133/5) which have different cell wall compositions (silica frustule, presence/absence of calcite and cellulose plates), we demonstrate that INT does not have a toxic effect on oxygen consumption at short incubation times. There was no difference in the oxygen consumption of a culture to which INT had been added and that of a replicate culture without INT, for periods of time ranging from 1 to 7 hours. For four of the cultures (*T. pseudonana* CCMP1080/5, *P. carterae* PLY-406, *E. huxleyi* RCC1217, and *O. marina* CCMP1133/5) the log of the rates of dissolved oxygen consumption were linearly related to the log of the rates of INT reduction, and there was no significant difference between the regression lines for each culture (ANCOVA test, F = 1.696, df = 3, p = 0.18). Thus, INT reduction is not affected by the structure of the plankton cell wall and a single INT reduction to oxygen consumption conversion equation is appropriate for this range of eukaryotic plankton. These results further support the use of the INT technique as a valid proxy for marine plankton respiration.

**Data Availability Statement:** All relevant data is available in the British Oceanographic Data Centre repository centre (www.bodc.ac.uk). The DOI is:

doi:10.5285/9727fcab-8aec-3fea-e053-
6c86abc05dd9 With a short doi: doi:10/dd6z

**Funding:** This study was supported by funds from
The Leverhulme Trust, RPG-2017-089 awarded to
CR (https://www.leverhulme.ac.uk/) and the
Natural Environment Research Council (NERC) of
the United Kingdom, NE/R000956/1 awarded to CR
(https://nerc.ukri.org/). The funders had no role in
study design, data collection and analysis, decision
to publish, or preparation of the manuscript.

**Competing interests:** The authors have declared
that no competing interests exist.

## Introduction

The reduction of tetrazolium salts has been widely applied to indicate the viability of prokaryotic and eukaryotic cells [1, 2], as well as in assays to measure cellular electron transport system activity [3–5]. Several studies have proposed that the reduction of the 2-para (iodophenyl)-3 (nitrophenyl)-5(phenyl) tetrazolium chloride (INT) to its reduced form, formazan ($INT_f$), could be used as a proxy for the respiratory activity of plankton cells [6, 7]. The increased sensitivity of this measurement, allowing less disruption of the sample and shorter incubation times (from minutes to hours) compared to the traditional Winkler titration method (usually 24 hours), has enabled improved understanding of the spatial and temporal variability of plankton respiration [8] as well as the apportionment of total plankton respiration to different size classes [8–10]. However, despite validation of INT reduction ($INT_R$) as a proxy for plankton respiration [6], the method remains controversial [11, 12]. Several researchers have shown that the reduction of soluble tetrazolium salts can have a toxic effect on cells, potentially due to formazan production inhibiting the electron transport system [13], leading to a reduction in metabolic activity [5, 12]. However, this reduction in metabolism is not immediate, and therefore there is a period of time when the toxic effects are not measurable [12, 14]. As far as we are aware, there are only three studies that have determined the effect of INT on cellular oxygen consumption, one with groundwater isolates [5], one with soil samples [14], and one with cultures of a marine haptophyte and a marine bacterium [12]. Results from these studies showed similar oxygen consumption in samples amended with INT compared to controls for periods of time ranging from 1 to 4 hours [12, 14], and noted a toxic effect thereafter. However, it is difficult to extrapolate the observations of toxicity in a single marine haptophyte [12] to all marine eukaryotes. Therefore, more information on any toxic effect of INT on oxygen consumption in a range of cultured marine eukaryotes would indicate any possible bias of the INT reduction method in natural plankton samples.

INT is a positively charged and cell permeable cation [15] which, in theory, freely passes through the cell walls of eukaryotes, and once inside the cell, is reduced to its formazan salt in the mitochondria [15, 16]. Plankton encompass a diverse group of organisms with a range of different cell wall compositions. Eukaryotic plankton cell walls can be made of silica frustules (diatoms), plates of calcite (coccolithophorids) or cellulose thecal plates (dinoflagellates). One of the assumptions of the INT reduction method is that the rate of reduction is consistent between organisms, independent of the cell wall structure. However, cell walls can act as a physical barrier to the diffusion of several molecules and metals into the cell [17, 18].Therefore, in order to continue to use a single relationship to convert INT reduction to oxygen consumption ($CR_{O_2}$) [6] for all eukaryotic plankton it is important to test this assumption with representative organisms covering the range of cell wall compositions. In this study, we tested the hypothesis that not all eukaryotic plankton cells take up and reduce INT at the same rate, and therefore there is not only one relationship between INT reduction and oxygen consumption, but several, depending on the cell wall structure of the organism (Fig 1).

## Material and methods

### Cultures and growth conditions

We chose 5 species from four dominant marine plankton groups with different cell wall characteristics: one diatom *Thalassiosira pseudonana* CCMP1080/5, with a silica cell wall; two coccolithophorides: *Pleurochrysis carterae* PLY-406 that produces calcite scales, and *Emiliania huxleyi* RCC1217 a non-calcifying strain; and two dinoflagellates: *Scrippsiella* sp. RCC1720 which has a cell wall composed of vesicles with cellulose thecal plates, and *Oxyrrhis marina*

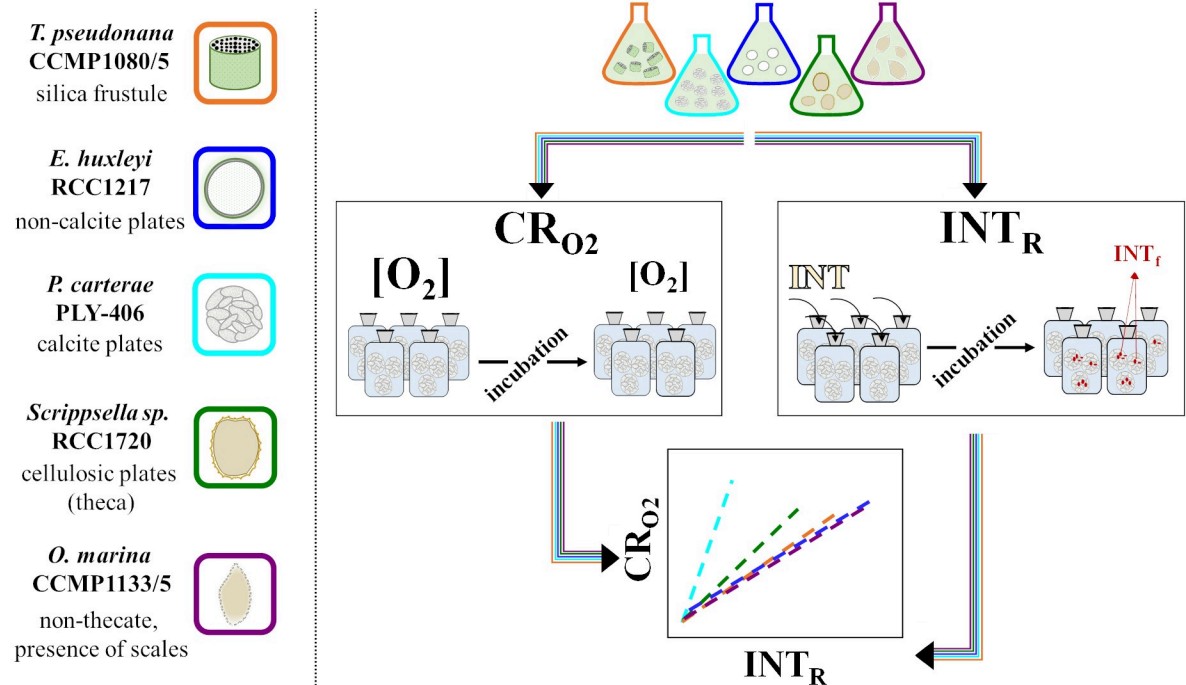

**Fig 1. Schematic diagram of the working hypothesis.** Oxygen consumption ($CR_{O2}$), oxygen concentration $[O_2]$, INT reduction ($INT_R$) and formazan ($INT_f$).

CCMP1133/5 a non-thecate heterotrophic dinoflagellate. *Thalassiosira pseudonana* CCMP1080/5 and *Oxyrrhis marina* CCMP1133/5 cultures were obtained from the Culture Collection of Marine Phytoplankton, currently the National Center for Marine Algae and Microbiota (NCMA, USA). *Emiliania huxleyi* RCC1217 and *Scrippsiella* sp. RCC1720 cultures were obtained from The Roscoff Culture Collection (RCC, France) and *Pleurochrysis carterae* PLY-406 from the Marine Biological Association (MBA Collection, UK). Cultures were maintained in the laboratory by regular subculturing. All eukaryotic cultures were grown in borosilicate glass flasks in 1 L of media (f/2-Si or K/2-Si [19–21]). The different media were prepared with autoclaved filtered (0.2 μm, Nucleopore polycarbonate membrane) seawater collected from the continuous filtered seawater supply system available in the laboratories of the Centre for Environment Fisheries and Aquaculture Science (Lowestoft, UK) pumped from the North Sea. The media used for the experiments were prepared at least 3–5 days in advance, stored in the dark at 4°C and acclimatized to the experimental temperature for at least 24 hours.

Cultures were grown under 140–190 μmol m$^{-2}$ s$^{-1}$ cool-white illumination in a 14:10 light: dark cycle at 12°C for *O. marina* CCMP1133/5 and at 20°C for the other cultures. *O. marina* CCMP1133/5 was fed on 0.02% (volume:volume) of *Chlorella vulgaris* CCAP211/11B. Cultures were grown for 4–8 days prior to experiments. Cell numbers and photosynthetic efficiency of PSII (Fv/Fm ratio) were measured daily to determine the growth and the health of the cultures. Cell numbers were determined by hemocytometer counts or with a Multisizer 4e Coulter Counter with a 100 μm aperture tube (Coulter Electronics, Ltd., Luton, UK.). The Fv/Fm ratio of the cultures was measured with a Walz PHYTO-PAM Phytoplankton Analyzer fluorometer (Walz, Effelrich, Germany) after cultures were kept for 10 minutes in the dark. The pulse amplitude modulated fluorescence measurements were completed at high repetition rates at four excitation wavelengths (470, 520, 645 and 665 nm). The intensity of the measuring light was 64 μmol quanta m$^{-2}$ s$^{-1}$, and the saturating pulse was supplied at the maximum level

specified by the Phyto-PAM (intensity level 10) at a pulse of 500 ms. All experiments were performed with cultures growing in exponential phase.

## Experimental design

The experiments ran for two consecutive days. The cultures were subsampled for the experiments at the end of the dark cycle on both days and dark conditions were maintained during the manipulation of the samples. On the first day a time series incubation was performed in order to determine the time at which the rate of formazan production began to decrease (see below), which is considered the maximum INT incubation time. INT incubation times equal to or shorter than this maximum were used on the second day during experiments to determine the relationship between oxygen consumption ($CR_{O2}$) and INT reduction ($INT_R$).

**Day 1: INT maximum incubation time experiment.** The cell abundance of the stock culture was measured one hour before starting the experiment (Table 1). A specific volume from the culture stock was harvested and resuspended in 1–2 L of fresh medium to obtain the desired final cell density (Table 1), and left for ~30 minutes in the dark prior to the time series experiment. Thirty mL samples were then pipetted from this sub-culture into 27–33 50 mL dark borosilicate glass bottles. The samples in three of these bottles were fixed with formaldehyde (2% w/v final concentration) and used as controls. Twenty minutes later all samples were inoculated with 0.75 mL of a 7.9 mM INT stock solution to achieve a final INT concentration of 0.2 mM. The bottles were then placed into temperature-controlled water baths, and maintained at the culturing temperature (temperature variability ± 0.4 ˚C of the culturing temperature) (Table 1) for up to 22 hours depending on the cultured organism and the expected INT reduction rate. At each of 8 to 10 time points during this incubation, triplicate samples were fixed with formaldehyde as in the controls described above. The INT reduced at each time point was determined as described below. The maximum incubation time was considered to be the time at which the rate of formazan production began to decrease. Up until this point, the relationship between the concentration of formazan and time remained linear. Due to the complexity of the experiments and the time necessary to perform a single maximum incubation time experiment (12–14 h), these experiments had no biological replication.

Formazan concentration undergoes an exponential increase that can be fitted to a 3-parameter 'exponential increase to a limit' function ($INT_f = yo + a * (1 - e^{(-b*t)})$, where yo, a and b are the parameters of the equation, and t is the incubation time). The approach used to estimate the time when the rate of formazan production began to decrease was as follows. We compared the $INT_f$ concentration estimated applying the 'exponential increase to a limit' function with the $INT_f$ concentration estimated from the linear regression equation obtained with the first three values of the relationship between the $INT_f$ concentration and incubation time. When the linear regression had a coefficient of determination smaller than 0.95, we used the first four data points in order to improve the prediction of the regression. As the linear regression with the first 3 or 4 data points is subject to error, we calculated a confidence region delimited by the slope ± one third of the standard error of the slope. This confidence region was chosen instead of the 95% confidence interval of the linear regression, as the 95% confidence interval was unacceptably wide due to the low number of datapoints (n = 3 or 4). The confidence region proposed here is 5 times more conservative than the 95% confidence interval of the linear regression. The time at which the $INT_f$ concentration estimated applying the 'exponential increase to a limit' function diverged from the confidence region of the $INT_f$ concentration calculated by linear regression was taken as the time when the rate of formazan production decreased with time (Fig 2).

**Table 1. Summary of the temperature, cell abundance, maximum incubation time and the estimated oxygen consumed at the maximum incubation time for the different cultures during the INT time series incubations.**

| Species | Date | Temperature | Cell abundance | Maximum incubation time | O₂ consumed by the maximum incubation time |
|---------|------|-------------|----------------|------------------------|--------------------------------------------|
| | | (˚C) | (cells mL⁻¹) | (h) | (µmol L⁻¹) |
| *T. pseudonana* | 04/12/2017 | 18.8 | 18000 | 1.82 | 1.01 |
| *T. pseudonana* | 23/10/2018 | 19.5 | 43000 | 1.38 | 0.97 |
| *T. pseudonana* | 23/10/2018 | 19.5 | 66000 | 1.08 | 1.20 |
| *T. pseudonana* | 13/11/2018 | 21.1 | 69000 | 0.26 | 0.44 |
| *E. huxleyi* | 05/02/2018 | 18.0 | 16000 | 2.73 | 0.63 |
| *E. huxleyi* | 26/02/2018 | 17.9 | 13200 | 0.97 | 0.02 |
| *E. huxleyi* | 12/03/2018 | 19.4 | 12000 | 2.92 | 0.69 |
| *Scrippsiella* sp. | 27/03/2018 | 19.8 | 200 | 0.92 | 0.88 |
| *Scrippsiella* sp. | 09/04/2018 | 19.8 | 250 | 0.4 | 0.50 |
| *O. marina* | 16/12/2018 | 12.5 | 1800 | 3.8 | 1.04 |
| *O. marina* | 12/01/2019 | 11.8 | 1000 | 3.9 | 1.34 |
| *P. carterae* | 11/03/2019 | 18.2 | 4500 | 0.38 | 0.37 |
| *P. carterae* | 19/03/2019 | 19.3 | 1700 | 0.48 | 0.35 |

Concurrent changes in oxygen concentration in cultures with and without INT addition were determined by continuously monitoring the oxygen concentration using optical probe sensors (FOSPOR-R probe, Ocean Optics) in 5 INT time series experiments (2 with *T. pseudonana* CCMP1080/5, 2 with *P. carterae* PLY-406 and 1 with *O. marina* CCMP1133/5). Unfortunately, there are no oxygen optode data available for the *E. huxleyi* RCC1217 experiments, as a chemical reaction between the metallic optode needle and the medium precluded accurate measurements. The optodes were re-coated with a non-reactive film for further experiments. Fospor-R optodes have an integrated multipoint calibration performed by the manufacturer. Prior to the experiments, the appropriate isothermal calibration curve was selected followed by a single point calibration with air-saturated and temperature equilibrated media, following the manufacturer's instructions. Three ~50 mL borosilicate bottles, each containing an optical probe, were used. The first bottle was filled with medium and used as a control to quantify the changes in oxygen concentration due to changes in the temperature of the water bath (< ±0.4 ˚C). The second bottle was filled with culture and used to measure the respiration rate of the

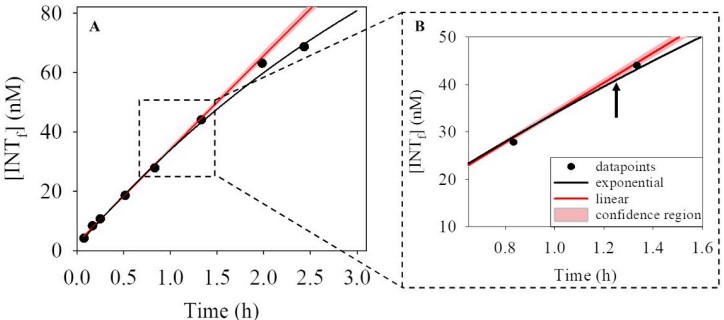

**Fig 2. Example of the procedure used to calculate the maximum incubation time for the INT reduction method.** A) INT formazan (INT_f) concentration at the incubation times (black dots), the fitted 'exponential increase to a limit' line (black line), the linear regression calculated with the first three data points (red line) and the confidence region (area within ± one third of the standard error of the linear slope, red shaded area). B) Zoom-in of the dashed rectangle in A. The arrow indicates the time when the fitted 'exponential increase to a limit' function diverges from the confidence region of the linear regression.

culture, and the third bottle was filled with culture to which INT was added in order to measure any change in oxygen consumption due to the addition of INT. The medium, culture and culture+INT samples were incubated in darkness in a water bath at the culturing temperature. The optical probes were tightly inserted into the bottles such that there was no interchange of oxygen between the medium/culture inside the bottle and the water from the incubator bath. Oxygen concentration and temperature were semi-continuously recorded under an on:off cycle of 20:40 seconds for at least 4 hours.

**Day 2: Oxygen consumption–INT reduction relationship experiments.** For each $CR_{O2}$-$INT_R$ experiment, we set up 3–4 containers (5 L borosilicate glass flask or a 10 L Thermo Scientific I-Che LDPE Cubitainer). One of the containers was filled with medium which was used as a control to check if there was any oxygen consumption or INT reduction by the medium. The cell abundance of the stock culture was measured before adding a specific volume to medium in the remaining 2–3 containers in order to achieve the required cell density (see Table 2 for final cell concentration). In 4 experiments the culture containers were duplicate or triplicate cultures with similar cell abundance (1 experiment with *T. pseudonana* CCMP1080/5, 2 with *E. huxleyi* RCC1217, and 1 with *Scrippsiella* sp. RCC1720), while in 8 experiments the culture containers had different cell abundances of the same culture (2 experiments with *T. pseudonana* CCMP1080/5, 2 with *P. carterae* PLY-406, 1 with *E. huxleyi* RCC1217, 1 with *Scrippsiella* sp. RCC1720, and 2 with *O. marina* CCMP1133/5). All containers (medium control and cultures) were stirred (250 rpm) while incubated in a temperature controlled water bath (temperature controlled room for *O. marina* CCMP1133/5) in the dark. The incubation temperatures were within ± 0.4°C of the culturing temperature (Table 2). The containers were sampled at two time points, first at 30 minutes after the addition of the stock culture, and then again after 3–4 hours.

## INT reduction

From each treatment (medium control or culture), at each of these two sampling time points, five 30 mL samples were collected into 50 mL dark borosilicate glass bottles. Two of these samples were fixed by adding formaldehyde (2% w/v final concentration) and used as controls. The 2 controls and 3 samples were then inoculated with a sterile solution of 7.9 mM INT solution to give a final concentration of 0.2 mM and incubated in the same temperature controlled water bath as the dissolved oxygen samples (see below) for a time equal to or shorter than the maximum incubation time determined the previous day. Incubations were terminated by adding formaldehyde to the remaining three samples, and together with the 2 controls, they were filtered onto 0.2 μm pore size polycarbonate filters, air-dried, and stored frozen. The INT reduced formazan was extracted with propanol following García-Martín et al. [6] and the absorbance at 485 nm determined using a Perkin Lambda 25 spectrophotometer. Hourly INT reduction for each treatment was calculated as the average INT formazan in the incubated samples (n = 3) minus the average INT formazan in the controls (n = 2) divided by the incubation time.

## Dissolved oxygen consumption

Respiration was determined as the decrease in dissolved oxygen after a dark incubation. The incubation time was as short as possible, and therefore varied from experiment to experiment, depending on the density of the culture. At each of the two sampling time points, samples from each medium control or culture were carefully siphoned into ten gravimetrically calibrated 60 mL borosilicate glass bottles. Five bottles were fixed at the start of the incubation ("zero") with 0.5 mL of 3 M manganese sulphate and 0.5mL of 4 M sodium iodide/8 M sodium hydroxide solution [22]. The other five bottles were placed underwater in darkened

**Table 2. Summary of the species, date, number of culture containers, temperature (T), cell abundance and incubation time for the INT reduction (INT$_R$) and dissolved oxygen consumption (CR$_{O2}$) techniques for the different cultures during the CR$_{O2}$-INT$_R$ experiments.**

| Species | Date | T | Cell abundance | Incubation time | |
|---|---|---|---|---|---|
| | | | INT$_R$ | CR$_{O2}$ | |
| | | (°C) | (cells mL$^{-1}$) | (h) | (h) |
| T. pseudonana | 05/12/2017 | 18.8 | 13000–18000 | 0.10 | 5.5 |
| T. pseudonana | 24/10/2018 | 19.5 | 35000–43000 | 0.63 | 3.1 |
| T. pseudonana | 14/11/2018 | 21.1 | 27000–40000 | 0.30 | 5.7 |
| E. huxleyi | 07/02/2018 | 18.0 | 10000–13000 | 1.50 | 24.0 |
| E. huxleyi | 27/02/2018 | 17.9 | 10000–12000 | 1.50 | 24.0 |
| E. huxleyi | 13/03/2018 | 19.4 | 12000 | 0.87 | 23.2 |
| Scrippsiella sp. | 28/03/2018 | 19.8 | 213.00 | 1.00 | 3.0 |
| Scrippsiella sp. | 10/04/2018 | 19.8 | 140–250 | 0.23–0.27 | 2.5 |
| O. marina | 17/12/2018 | 12.5 | 1500–1750 | 3.18 | 24.4 |
| O. marina | 14/01/2019 | 11.8 | 500–1000 | 1.98 | 22.3 |
| P. carterae | 12/03/2019 | 18.2 | 3200–6000 | 0.25 | 5.3 |
| P. carterae | 20/03/2019 | 19.3 | 1000–1700 | 0.23–0.30 | 4.0 |
| P. carterae | 23/03/2019 | 19.3 | 667 | 0.33 | 3.7 |
| P. carterae | 26/03/2019 | 16.8 | 3300–4580 | 0.25 | 4.5 |

temperature-controlled incubators for a specific incubation time ("dark", see Table 2 for the incubation time). "Dark" bottles were fixed as described for the "zero" bottles after the incubation time. Dissolved oxygen concentration was measured by automated Winkler titration performed with a Metrohm 765 burette to a photometric end point [22]. Hourly respiration was calculated from the difference in oxygen concentration between the mean of the replicate "zero" measurements (n = 5) and the mean of the replicate "dark" measurements (n = 5) divided by the incubation time. The respiration per cell (CR$_{O2\_cell}$) was calculated as the respiration of the culture divided by the cell abundance of that culture.

The estimation of respiration from a two point ('start' and 'final') incubation relies on the assumption of a linear decrease in dissolved oxygen. In order to confirm this assumption, we measured the oxygen concentration semi-continuously by means of optical oxygen probes in one of the culture containers in 8 CR$_{O2}$-INT$_R$ experiments (2 with *T. pseudonana* CCMP1080/5, 2 with *Scrippsiella* sp. RCC1720, 2 with *O. marina* CCMP1133/5 and 2 with *P. carterae* PLY-406). Oxygen consumption was calculated as the slope of oxygen concentration measured with the optical sensors versus time.

The incubations for oxygen consumption measured with optodes (<24 h) were typically much longer than those for oxygen consumption measured with Winkler titrations (usually ~ 5 h), so that although oxygen consumption varied over the full length of the optode incubation, linearity was achieved for the length of time that the oxygen bottles were incubated (S1 Fig). In fact, oxygen consumption measured semi-continuously (optodes) and discretely (Winkler titration) were significantly correlated (r = 0.57, p = 0.048, n = 8) and not significantly different (paired t-test, t = 0.6.86, df = 7, p = 0.515). Thus, we have assumed that the decrease in oxygen was linear in all incubated samples and that the oxygen consumption derived from the two point Winkler titration method was not biased.

## Data analysis

To ascertain whether cell number, the respiration per cell, or the oxygen consumption of the sample (a combination of cell number and respiration per cell, see below) had the greatest

influence on the maximum INT incubation time, we performed a Tau-Kendall non-parametric correlation between them. The oxygen consumed at the maximum incubation time ($O_{2[MIT]}$) on the Day 1 experiment was calculated according to the following formula:

$$O_{2[MIT]} = (CR_{O2\_cell}) * A_1 * MIT,$$

where $CR_{O2\_cell}$ is the respiration per cell from the Day 2 experiment, $A_1$ is the cell abundance from the Day 1 experiment and MIT is the maximum incubation time from the Day 1 experiment.

The correlation between oxygen consumption measured with optical sensors and by Winkler titration was analysed by Tau-Kendall non-parametric correlation, and the possible differences were analysed by paired t-test analysis.

The $CR_{O2}$-$INT_R$ relationships were determined with a type I ordinary least squares model regression after the data were log-transformed. The implicit assumption of our hypothesis (Fig 1) is that $INT_R$ must be significantly related to $CR_{O2}$. If this assumption was not met for an organism (an insignificant $CR_{O2}$-$INT_R$ relationship), the data were not included in the analysis. An analysis of covariance (ANCOVA) was used to compare the $CR_{O2}$-$INT_R$ regression lines obtained from each plankton culture.

The combined $CR_{O2}$-$INT_R$ relationship of the cultures was compared with the published relationship derived from natural plankton populations [6] using a Clarke test [23].

## Results

### Maximum incubation time for the INT

The maximum incubation time (considered as the maximum time that the rate of formazan production remained constant) was different depending on the species tested and there were differences within species (Table 1). This ranged from 0.26 h for *T. pseudonana* CCMP1080/5 (cell density 69000 cells mL$^{-1}$) to 3.9 h for *O. marina* CCMP1133/5 (cell density 1000 cells mL$^{-1}$) (Table 1). There was a significant correlation between the maximum incubation time and the calculated $O_{2[MIT]}$ (r = 0.49, p = 0.02, n = 13), but not with the cell abundance (r = -0.12, p = 0.542, n = 13) or the respiration per cell (r = -0.09, p = 0.669, n = 13). These results suggest that the INT maximum incubation time is influenced by the respiration of the sample, which is a combination of cell abundance and respiration per cell.

### Effect of INT on the oxygen consumption rates

The continuous monitoring of oxygen concentration with optical sensors allowed us to test the effect of INT on oxygen consumption. Despite the different oxygen consumption trends and rates, our results showed that in all of our experiments, there was no difference in the oxygen consumption of a culture to which INT had been added compared to a control culture without addition of INT during the time when the formation of formazan remained linear (Fig 3). At longer incubation times the effect of the INT on oxygen consumption followed a similar pattern: the rate of oxygen consumption was lower in samples to which INT had been added compared to the samples without INT (Figs 3 and S1). The time at which the respiration rate of samples with and without INT began to differ, varied between experiments. The toxicity (or difference in the oxygen consumption of the two samples) was evident at incubation times shorter than one hour in a *T. pseudonana* CCMP1080/5 experiment, which had the highest measured respiration rates (~2.5 μmol O$_2$ L$^{-1}$ h$^{-1}$, Fig 3A), while the toxicity was not detectable until >7 hours in the INT treated sample in four experiments (*O. marina* CCMP1133/5 Fig 3C; *T. pseudonana* CCMP1080/5, Figure B in S1 Fig; *P. carterae* PLY-406, Figure E-F in S1

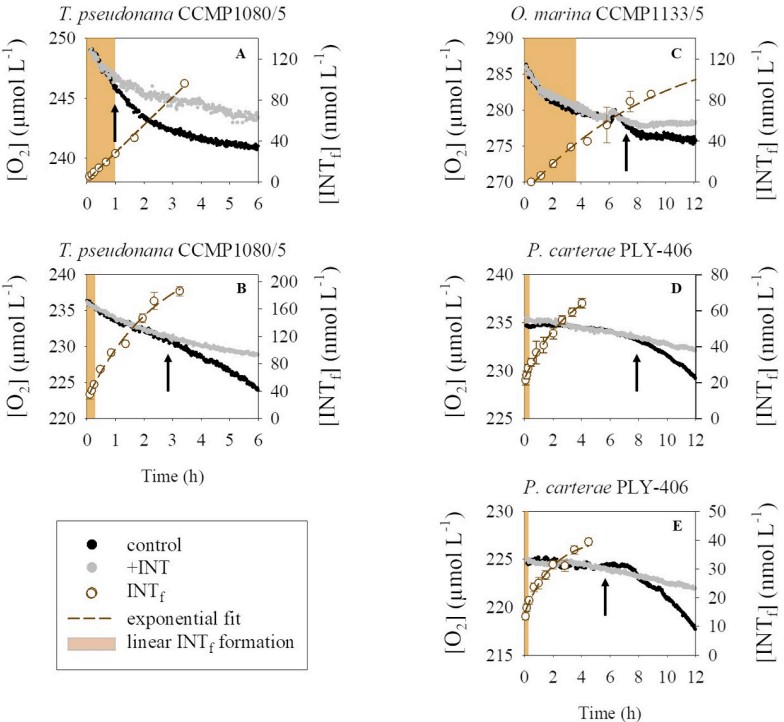

**Fig 3.** Effect of INT on the oxygen consumption of *Thalassiosira pseudonana* CCMP1080/5 (A, B), *Oxyrrhis marina* CCMP1133/5 (C) and *Pleurochrysis carterae* PLY-406 (D, E) during time series incubations in the dark. The oxygen concentration (left hand y axis) of samples without INT (controls) is shown in black, and that of samples after addition of INT (final concentration 0.2 mM) in grey. The concentration of formazan is shown as open circles ± standard error on the right-hand y axis with the fitted exponential curve (dashed brown line). The time at which the decline in oxygen diverges between the INT treated samples and the controls is shown as a thick black arrow, and the time during which the formazan formation is linear is shown as a pale brown shaded area. Note that oxygen concentration, incubation time and formazan concentration axes are different for each experiment.

Fig), which had much lower respiration rates ($<1$ µmol $O_2$ $L^{-1}$ $h^{-1}$). These results support our previous results that formazan production, and subsequent accumulation and toxicity, are related to the respiration of the sample.

## Effect of the cell wall on the oxygen consumption-INT reduction relationship

Overall, there was no oxygen consumption or INT reduction in the media controls as the 95% confidence interval included the null value (data available from the British Oceanographic Data Centre repository, www.bodc.ac.uk, short DOI doi:10/dd6z). This reassured us that the media were free of respiring organisms, and did not have the capacity to reduce INT, and therefore that the media did not contribute to the $CR_{O2}$-$INT_R$ relationship.

All cultures tested showed a significant relationship ($p < 0.05$) between the oxygen consumed and the INT reduced, except *Scrippsiella* sp. RCC1720 ($p > 0.05$) (Fig 4). Excluding *Scrippsiella* sp. RCC1720 from the analysis, there was no difference in the $CR_{O2}$-$INT_R$ regression lines between the different species (ANCOVA test, F = 1.696, df = 3, p = 0.18). As there was no significant difference between the different cultures, we grouped all the culture data (except *Scrippsiella sp*. RCC1720) and calculated a single $INT_R$ to $CR_{O2}$ conversion equation. The calculated conversion equation with data from the four organisms is: $\log CR_{O2} = 0.61\log INT_R + 0.62$, $R^2 = 0.87$, n = 49, $p < 0.001$. Despite the lack of relationship between the

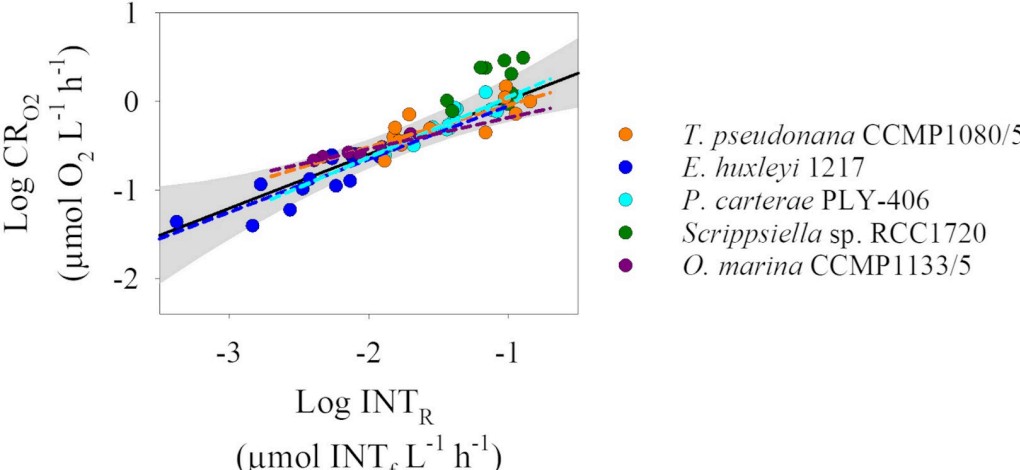

**Fig 4. Log-log relationship between dissolved oxygen consumption (CR$_{O2}$) and INT reduction (INT$_R$) for the different cultures tested.** *Thalassiosira pseudonana* CCMP1080/5 in orange (n = 12), *Emiliania huxleyi* RCC1217 in blue (n = 13), *Pleurochrysis carterae* PLY-406 in cyan (n = 15), *Scrippsiella sp*. RCC1720 in green (n = 9) and *Oxyrrhis marina* CCMP1133/5 in purple (n = 10). The coloured dashed lines represent the Type I linear regression line for each culture. Note that *Scrippsiella sp*. RCC1720 does not have a regression line as the relationship between CR$_{O2}$ and INT$_R$ was not significant (p >0.05). The grey shaded area represents the 95% confidence interval for the linear regression (black line) obtained with all the data except *Scrippsiella* sp. RCC1720.

CR$_{O2}$-INT$_R$ paired data from *Scrippsiella sp*. RCC1720, 4 out of 9 of the data points from *Scrippsiella sp*. RCC1720 are within the 95% confidence interval of the regression line from the grouped culture data (Fig 4).

In order to test the applicability of the results from our culture experiments to natural populations, we compared the results from the analysis performed in the previous section with the most recently published CR$_{O2}$-INT$_R$ database for natural plankton populations [6]. There are differences in the methodology used between culture experiments and natural populations including the incubation time for the oxygen consumption technique. While for natural populations, oxygen consumption was measured over 24 hours, in the culture experiments the

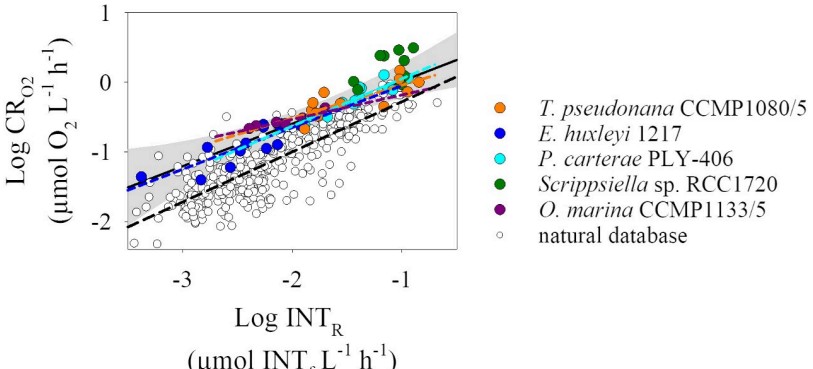

**Fig 5. Log-log linear regressions between dissolved oxygen consumption (CR$_{O2}$) and INT reduction (INT$_R$).** Coloured circles correspond to the cultured data and white circles correspond to the natural plankton database from [6]. Black solid and dashed lines represent the type I linear regressions for the cultured eukaryotes and natural database (logCR$_{O2}$ = 0.72logINT$_R$ + 0.44 [6]), respectively. The grey shaded area represents the 95% confidence interval for the linear regression obtained with all the culture data except *Scrippsiella sp*. RCC1720.

incubation time for oxygen consumption was usually shorter than 24 hours due to the higher respiration rates of the cultures (see Table 2 for incubation times).

The rates of INT reduction measured in the culture experiments were within the range of variability measured in natural populations ($0.0003–0.021$ µmol $INT_f L^{-1} h^{-1}$). In addition, the rates of oxygen consumption in the cultures were generally within the upper limit of the range of variability measured in natural populations ($0.004–0.94$ µmol $O_2 L^{-1} h^{-1}$), except the oxygen consumption rates of *Scrippsiella sp*. RCC1720 which were 2-fold greater than the maximum rate of oxygen consumption measured in natural populations (Fig 5). All data obtained in the culture experiments lay above the regression line reported for the natural database ($logCR_{O2} = 0.72logINT_R + 0.44$, $R^2 = 0.69$, n = 249, p <0.001, [6], Fig 5). In addition, the conversion equation obtained from the culture experiments is significantly different to that of the natural database (Clarke test, t = 2.65, df = 444, p = 0.008). Our results indicate that oxygen consumption would be overestimated, by 1.5 to 3.5 fold, if the $INT_R$ to $CR_{O2}$ conversion equation derived from the culture experiments was used to estimate oxygen consumption from INT reduction of a natural population instead of the $INT_R$ to $CR_{O2}$ conversion equation derived from natural plankton.

## Discussion

Exposure to INT is ultimately toxic [12] as the formation of the insoluble formazan deposits can lead to a reduction in the metabolic activity, and thus in oxygen consumption [12]. Our results confirm the toxic effects of INT, but as previously shown [6], the toxicity is not evident at short incubation times. The results from our culture experiments show, for the first time in several marine eukaryotes, that there was no difference between the oxygen consumption of a culture to which INT had been added and one without addition of INT during the time when the production of formazan remained linear. This observation was consistent for four different planktonic eukaryotic organisms (*Thalassiosira pseudonana* CCMP1080/5, *Pleurochrysis carterae* PLY-406, *Scrippsiella* sp. RCC1720 and *Oxyrrhis marina* CCMP1133/5) in the 11 experiments undertaken. The lack of difference in oxygen consumption between samples with and without addition of INT has only previously been reported for cultures of a marine haptophyte and a marine bacterium during ~1 hour of incubation [12] and in soil bacteria for ~4 hours [14]. Our results indicate that the incubation times used in our experiments, which were determined from time series incubations, were appropriate and that INT was not inhibiting the respiratory activity of the cells. Therefore, reinforcing previous work [6], the results show that INT reduction can be used as a proxy for plankton respiration when incubated for specifically determined short time periods. However, we cannot recommend a particular maximum incubation time and do not support the idea of a constant incubation time for the INT reduction method with natural plankton or eukaryotic cultures as other authors have suggested for prokaryotes [11]. The maximum incubation time (determined as the time after which INT had a negative effect on oxygen consumption) was not related to cell abundance or by implication to biomass alone. It was related to the respiratory activity of the sample (the combined effect of cell abundance and respiration per cell), being longer for samples with low respiration rates and shorter for samples with higher respiration rates.

Eukaryotic plankton encompass a diverse range of organisms with distinctly different cell wall structures. Some organisms possess a silica frustule (diatoms), others plates composed of calcite (coccolithophorids) or cellulose thecae (dinoflagellates). One of the main assumptions of the INT reduction technique is that INT penetrates the cell wall and is reduced at the mitochondria at the same rate for all organisms whatever the cell wall composition. There is no systematic study of the reduction of INT by marine eukaryotes, but the evidence from

prokaryotic studies [24, 25], and from the permeability of eukaryotic cell walls to different chemical compounds [18], and metals [17] challenge this assumption. However, we found no evidence to suggest that the characteristics of eukaryotic plankton cell walls have any effect on the capacity of INT to penetrate inside the cells. Experiments performed with four planktonic eukaryotic organisms characterized by different cell wall types showed similar oxygen consumption-INT reduction relationships. The only culture which did not show a significant relationship between INT reduction and oxygen consumption was *Scrippsiella* sp. RCC1720. This lack of relationship may simply be due to the small range of INT reduction and oxygen consumption rates measured in these culture experiments. However, ~50% of the paired $CR_{O2}$-$INT_R$ *Scrippsiella* sp. RCC1720 data are within the 95% confidence interval of the linear regression calculated for the rest of the cultures. Therefore, although we have shown its ability to reduce INT and the lack of any toxic effects at short incubation times, we recommend caution in using this eukaryote $CR_{O2}$-$INT_R$ conversion equation when working with thecate dinoflagellate cultures.

The $CR_{O2}$-$INT_R$ conversion equation obtained with the culture experiments differs from the most recent $CR_{O2}$-$INT_R$ conversion equation derived from natural plankton populations [6]. However, it may be inappropriate to compare these two $CR_{O2}$-$INT_R$ relationships as they are derived from different plankton populations: one is obtained from cultured planktonic eukaryotes growing exponentially in an enriched medium and the other is from natural plankton communities composed of a mixture of eukaryotes and prokaryotes at different stages of growth often under nutrient deplete conditions. In addition, differences could be due to biases in the INT reduction or oxygen consumption in the cultured experiments. On the one hand, INT reduction could have been influenced by the composition of the medium (f/2 or K/2). INT reduction of bacterial cultures is inhibited by metal ions (i.e. cobalt) [24], and can also be affected by pH and phosphate concentration [26, 27]. Our media contain trace metals, which may have interfered with the reduction of INT, resulting in a lower INT reduction rate in the cultured samples. On the other hand, the oxygen consumption of the cultures could be enhanced compared to natural plankton communities, as physiological rates (i.e. primary production and respiration) tend to be higher in cultured organisms than in natural populations [28, 29]. Such decreased INT reduction and/or increased oxygen consumption under culture conditions would result in the different culture versus natural population $CR_{O2}$-$INT_R$ relationships seen here. Studies on the relationship between plankton respiration (R), measured by oxygen consumption, and the activity of the electron transport system (ETS), report analogous differences in the cultured versus natural R/ETS ratio, with an even greater offset than the 1.5 to 3.5 fold differences observed in this study [30], and references therein]. Whatever the causes of the differences in the $CR_{O2}$-$INT_R$ conversion equation found in the culture experiments compared to natural populations, the future use of the INT reduction method to estimate plankton respiration should include whichever $CR_{O2}$:$INT_R$ conversion equation is most relevant to the community being studied.

Many phytoplankton organisms produce reactive oxygen species (ROS, e.g. $O_2^-$, $H_2O_2$), during aerobic metabolism [31]. These highly reactive compounds can act as reducing agents of tetrazolium salts [32]. The amount of ROS concentration inside the cells has been reported to increase in cultures under stress situations or nutrient deprivation conditions [32]. We did not measure the presence or the concentration of ROS in our experiments, and therefore we cannot infer whether there was any INT reduction by ROS species. The healthy and exponential growing conditions of our cultures suggest that the presence of ROS species inside the cells might have been low. We acknowledge that, to some degree, our INT reduction rates presented in this study and in García-Martín et al. [6] might be overestimated and we suggest caution in interpreting the results. However, due to the similar oxygen to INT relationships of the

different organisms tested, we consider that the effect of ROS species in our experiments and during the maximum INT incubation time might have been minimal or constant. Future studies should test the influence of reactive oxygen species on the reduction of INT to quantify this potential bias.

## Conclusions

This study validates our approach to determine the maximum incubation time as the time when the formazan formation ceases to be linear, confirms that the cells are not poisoned during our INT incubation times and contradicts the claim that INT toxicity invalidates the INT reduction method [12]. In addition, it demonstrates that the oxygen consumption–INT reduction relationship for eukaryotic plankton is not systematically affected by cell wall composition, and therefore, a single conversion equation between the moles of formazan produced and the moles of oxygen consumed can be used to estimate the respiration of natural populations of plankton.

## Supporting information

**S1 Fig.** Oxygen concentration over time measured in dark incubations of Thalassiosira pseudonana CCMP1080/5 (A, B), Scrippsiella sp. RCC1720 (C, D), Pleurochrysis carterae PLY-406 (E, F) and Oxyrrhis marina CCMP1133/5 (G, H) cultures during the CRO2-INTR experiments. The oxygen concentration of samples without INT (controls) is shown in black, and that of samples after addition of 0.2 mM INT in grey. Plots D and G do not have samples with INT added due to optode sensor failure. The incubation time for the oxygen incubations measured with Winkler titrations is shown as a light blue shaded area and the incubation time for the INT reduction method as a blue dashed box. Note the linear trend in the oxygen consumption during the incubation time for the Winkler discrete samples, and the lack of any difference between the oxygen consumption with and without addition of INT. Note that the oxygen concentrations and time intervals are different for each culture experiment.
(TIF)

## Acknowledgments

We thank Robert Utting for his help with the culturing and maintenance of the cultures, Erik Buitenhuis for advice with the different cultures and Stephen Dye for providing natural seawater to prepare our f/2-Si or K/2-Si media.

## Author Contributions

**Conceptualization:** E. Elena García-Martín, Carol Robinson.

**Data curation:** E. Elena García-Martín.

**Formal analysis:** E. Elena García-Martín, Isabel Seguro.

**Funding acquisition:** E. Elena García-Martín, Carol Robinson.

**Investigation:** E. Elena García-Martín, Isabel Seguro.

**Project administration:** Carol Robinson.

**Supervision:** Carol Robinson.

**Visualization:** E. Elena García-Martín.

**Writing – original draft:** E. Elena García-Martín.

**Writing – review & editing:** E. Elena García-Martín, Isabel Seguro, Carol Robinson.

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
