## [Decision Letter · Decision Letter 0]

16 Oct 2019

PONE-D-19-27198

INT reduction is a valid proxy for eukaryotic plankton respiration despite the inherent toxicity of INT and differences in cell wall structure

PLOS ONE

Dear Dr Garcia-Martin,

Thank you for submitting your manuscript to PLOS ONE. After careful consideration, we feel that it has merit but does not fully meet PLOS ONE’s publication criteria as it currently stands. Therefore, we invite you to submit a revised version of the manuscript that addresses the points raised during the review process.

We would appreciate receiving your revised manuscript by Nov 30 2019 11:59PM. To enhance the reproducibility of your results, we recommend that if applicable you deposit your laboratory protocols in protocols.io, where a protocol can be assigned its own identifier (DOI) such that it can be cited independently in the future. For instructions see: http://journals.plos.org/plosone/s/submission-guidelines#loc-laboratory-protocols

We look forward to receiving your revised manuscript.

Kind regards,

Antonietta Quigg, PhD

Academic Editor

PLOS ONE

Journal Requirements:

1.

2.  In order to enhance reproducibility, please clarify the origin of the strains used, and provide any further details available on the origin and composition of the seawater used.

3.  Our internal editors have looked over your manuscript and determined that it may be within the scope of our Life in Extreme Environments Call for Papers. The Collection will encompass a diverse range of research articles to better understand life and biogeochemistry in extreme environments. Additional information can be found on our announcement page: https://collections.plos.org/s/extreme-environments. If you would like your manuscript to be considered for this collection, please let us know in your cover letter and we will ensure that your paper is treated as if you were responding to this call. If you would prefer to remove your manuscript from collection consideration, please specify this in the cover letter.

Reviewers' comments:

Reviewer's Responses to Questions

**Comments to the Author**

1. Is the manuscript technically sound, and do the data support the conclusions?

Reviewer #1: Yes

Reviewer #2: Yes

Reviewer #3: Yes

2. Has the statistical analysis been performed appropriately and rigorously? 

Reviewer #1: Yes

Reviewer #2: Yes

Reviewer #3: Yes

3. Have the authors made all data underlying the findings in their manuscript fully available?

Reviewer #1: Yes

Reviewer #2: Yes

Reviewer #3: No

4. Is the manuscript presented in an intelligible fashion and written in standard English?

Reviewer #1: Yes

Reviewer #2: Yes

Reviewer #3: Yes

5. Review Comments to the Author

Reviewer #1: The work described in this manuscript addresses the important issue of the validity of the INT method to estimate respiration. The experiments were well planned, well conducted and well interpreted.

However, a number of points need attention.

Lines 32-33. A quantitative comparison is required of the times needed to get an estimate of respiration for the two methods.

Lines 37-40. Does this toxic effect include alteration to the function of the plasma membrane?

Line 44. ‘a marine bacterium’ (‘bacteria’ is the plural).

Line 51. A salt is, by definition, composed of a cation and an anion. Replace by ‘INT is a cell permeable cation [15]…..’.

Line 54. Prokaryotic plankton organisms, e.g. cyanobacteria, do have a cell wall, defined as structure outside the plasma membrane that can resist turgor (e.g. Walsby 1973 Limnology and Oceanography 18: 653-658; Ladas and Papageorgiou 2000 Photosynthesis Research 65: 155-164)

Lines 58-59. Clarify ‘intracellular diffusion’. To me, intracellular means ‘within the cell’. The implication here seems to be the diffusion into, not within, the cell.

Line 71. See my comment on Line 54. Are the cell coverings of the two coccolithophores and the two dinoflagellates turgor resistant? See Sikes and Wilbur 1982 Limnology and Oceanography 27: 18-26.

Line 80. ‘were’, not ‘was’; ‘media’ is plural.

Line 394. These references do not adequately distinguishing adsorptionon extracellular structures and entry into the cell. For the uptake of essential trace metal uptake into eukaryotic marine microalgal cells, there is a large body of information from the laboratory of F M M Morel in Princeton NJ. For the organic compound NADH, the permeability of the cell wall of chlorophytan freshwater microalgae depends on the presence or absence of a sporopollenin layer in the cell wall (Syrett and Thomas 1973 New Phytologist 72: 1307-1310).

Lines 396-398. Is there any mechanistic basis for the allocation of electrons between the reduction of INT and the reduction of oxygen?

Reviewer #2: In this paper the authors report on the use of the tetrazolium salt INT to determine respiration rates in microalgal cultures. The use of INT is well established as a mechanism to estimate electron transport activity but the present paper has the novelty of careful attention to the effects of time of incubation and potential toxicity on apparent rates of respiratory activity. As such it does represent a useful contribution to the literature. However there are a number of issues that require attention before the manuscript is suitable for publication.

line 29: I would hardly call use of INT 'recent' as its use as a measure of respiratory electron transport goes back a long way (Ted Packard proposed this in his 1971 paper as did Kenner, R.A. & Ahmed, S.I. Marine Biology (1975) 33: 119 for microalgae and Owens, T.G. & King, F.D. Mar. Biol. (1975) 30: 27. for zooplankton).

line 58: I am not convinced about the arguments that different cell wall structures represent a major barrier to the entry of INT. For metals at least the cell wall components act less as a permeability barrier than a binding site for ions . The plasmalemma would be more of a barrier to transport of a changed molecule such as INT into the cell.

line 79: Here and elsewhere 'media' is the plural of medium, so only use this (with 'were' ) if you used more than one type of medium

line 85: At what stage of growth were these cultures at the time of the experiments?This is important if experiments were done over 2 days as physiological characteristics can change rapidly in batch cultures. There is a brief reference to exponential stage cultures in the Discussion but this needs to be better defined early on.

lines 89/90: which type of PAM? Walz make lots of different forms of the instrument. More details such as saturating pulse intensity are needed

line 102-105: this looks like pseudoreplication rather than the use of independent biological replicates. Please clarify.

lines 122-123: Consider showing the equation for the 3-parameter ‘exponential increase to a limit’ function

lines 208-9: This is fair enough as an estimate of steady state dark respiration, but how does this relate to the fact that respiration rates immediately after a light exposure can be up to x10 those at steady state in the dark? See e.g. Beardall et al a994 Journal of Plankton Research Vol.16 no.10 pp. 1401 -1410. This is also relevant to the Discussion.

line 366: Stating this is 'for the first time' is not strictly accurate given the mention to reference 12 cited below (line 374)

line 386: choice of incubation time is further complicated by the enhanced post illumination respiration shown by many phytoplankton species (see comment above)

Reviewer #3: The manuscript "INT reduction is a valid proxy for eukaryotic plankton respiration despite the inherent toxicity of INT and differences in cell wall structure" focuses on an alternate method for measuring plankton respiration with the help of INT. Overall the manuscript is well written and has applicable objective, however the authors would need to address the following concerns:

1. Tetrazolium salts can also be reduced by reactive oxygen species in the cell, therefore if the plankton is under oxidative stress, use of this technique can lead to over-estimation of the respiratory rates. The authors need to either compare the reduction of INT in the presence and absence of oxidative stress or at least discuss the possibility of interference by ROS and advise caution with the interpretation.

2. Fig. 3 shows extreme variation in results even within the same species, for example the oxygen consumption patterns are very different between the two figures of T. pseudonana. This suggest extreme variability in this method, not just between species but within the same species. Can the authors provide an explanation for this?

Also, it looks like E. huxleyi and Scrippsella sp. were excluded from this measurement, is there a particular reason?

3. Measurement of respiration with the help of Clark-type oxygen electrode usually takes about 10-15 mins? How does the authors just the use of INT as a better way to measure respiration? Moreover, the authors themselves have used optodes to compare the INT measurements. What advantages does INT measurements have over clark type electrodes and optodes? Consider discussing this in terms of cost, time, and sample volume?

6. PLOS authors have the option to publish the peer review history of their article (what does this mean?). If published, this will include your full peer review and any attached files.

Reviewer #1: No

Reviewer #2: No

Reviewer #3: No

---

## [Author Response · Author response to Decision Letter 0]

31 Oct 2019

Journal Requirements:

 The manuscript meets PLOS ONE’s requirements

2. In order to enhance reproducibility, please clarify the origin of the strains used, and provide any further details available on the origin and composition of the seawater used.

More information on the origin of the strains and seawater are provided (lines 75-81 and lines 84-86).

3. Our internal editors have looked over your manuscript and determined that it may be within the scope of our Life in Extreme Environments Call for Papers. The Collection will encompass a diverse range of research articles to better understand life and biogeochemistry in extreme environments. Additional information can be found on our announcement page: https://collections.plos.org/s/extreme-environments. If you would like your manuscript to be considered for this collection, please let us know in your cover letter and we will ensure that your paper is treated as if you were responding to this call. If you would prefer to remove your manuscript from collection consideration, please specify this in the cover letter.

Thank you for the offer to include the manuscript in the “Life in Extreme Environments” Call. We consider that this study does not address any of the topics proposed in the call. Our study deals with eukaryotic plankton grown in healthy, nutrient rich, non hostile conditions. 

All data have been sent to BODC on the 28th October 2019. I have been notified that they will need two weeks to process the data and provide a DOI for it. I will forward it to the journal as soon as I have it. 

All data have been sent to BODC repository and a DOI number will be inserted upon reception

Response to Reviewers

We are happy to read that the reviewers share our opinion that the manuscript contributes to the existing literature. We would like to thank the three reviewers for their comments and suggestions for additional references, which has helped to improve the manuscript. All the points have been dealt with in the text and new references have been included. These changes are indicated in our responses below specifying the lines in the latest submitted manuscript (after track changes). We have also addressed additional journal requirements. 

Reviewer #1: The work described in this manuscript addresses the important issue of the validity of the INT method to estimate respiration. The experiments were well planned, well conducted and well interpreted. However, a number of points need attention.

Lines 32-33. A quantitative comparison is required of the times needed to get an estimate of respiration for the two methods.

We have indicated (lines 32-33) that the incubation time for the INT method is from minutes to hours, while for the Winkler titration method is usually 24 h. 

Lines 37-40. Does this toxic effect include alteration to the function of the plasma membrane? 

As far as we are aware, reports of the toxicity of INT and other tetrazolium salts, are based on the inhibition of bacterial growth (Coallier et al. 1994 Can J Microbiol 40:830-836, Hatzinger et al. 2003 J Microbiol Meth 52: 47– 58, Ullrich et al. 1996 Appl Environ Microbiol, 62: 4587–4593), inhibition of oxygen consumption (Villegas-Mendoza et al. 2015, Ullrich et el. 1996), and inhibition of other metabolic activities (Ullrich et al. 1996). There is no mention in the literature of any alteration to the function of the plasma membrane. 

Line 44. ‘a marine bacterium’ (‘bacteria’ is the plural). Corrected (line 43). 

Line 51. A salt is, by definition, composed of a cation and an anion. Replace by ‘INT is a cell permeable cation [15]…..’. Corrected (line 50).

Line 54. Prokaryotic plankton organisms, e.g. cyanobacteria, do have a cell wall, defined as structure outside the plasma membrane that can resist turgor (e.g. Walsby 1973 Limnology and Oceanography 18: 653-658; Ladas and Papageorgiou 2000 Photosynthesis Research 65: 155-164). 

Corrected. We have changed this sentence to focus only on the diversity of eukaryotic plankton cell wall composition (line 53). 

Lines 58-59. Clarify ‘intracellular diffusion’. To me, intracellular means ‘within the cell’. The implication here seems to be the diffusion into, not within, the cell. 

We have modified the sentence to indicate that the diffusion is into the cells (line 57). 

Line 71. See my comment on Line 54. Are the cell coverings of the two coccolithophores and the two dinoflagellates turgor resistant? See Sikes and Wilbur 1982 Limnology and Oceanography 27: 18-26. 

Unfortunately, we cannot comment on the turgor resistance of the species. We did not check the turgor pressure or the osmotic properties of any of the cultures, so we cannot respond to the reviewer with any certainty. From our results with the calcified and noncalcified organisms, we can indicate that the coccosphere of the calcified organisms seems to not affect diffusion into the cell, as the relationship between oxygen consumption and INT reduction was similar for the two species. If the coccosphere were negatively affecting diffusion into the cell, then there should be a different slope between the CRO2 and INTR, assuming that once inside the cell, the INT has a similar behaviour. The study of the similar or different CRO2 – INTR relationship is one of the main objectives of this study as it will indicate whether a single moles of INT to moles of oxygen conversion equation is adequate for all natural plankton populations which encompass a mixture of different eukaryotic organisms. 

Line 80. ‘were’, not ‘was’; ‘media’ is plural. 

Corrected in several part of the manuscript (lines 112, 167, 171, 174, 202, 427).

Line 394. These references do not adequately distinguishing adsorptionon extracellular structures and entry into the cell. For the uptake of essential trace metal uptake into eukaryotic marine microalgal cells, there is a large body of information from the laboratory of F M M Morel in Princeton NJ. For the organic compound NADH, the permeability of the cell wall of chlorophytan freshwater microalgae depends on the presence or absence of a sporopollenin layer in the cell wall (Syrett and Thomas 1973 New Phytologist 72: 1307-1310). 

Thank you for the reference suggested. We now refer to these studies as the reviewer suggests (lines 57 and 406).

Lines 396-398. Is there any mechanistic basis for the allocation of electrons between the reduction of INT and the reduction of oxygen? 

Four electrons are needed to reduce one mol of oxygen while only 2 electrons are needed for a mol of INT, but we are not aware of any mechanistic basis for the allocation of electrons between the INT and the oxygen. We are currently studying the enzymes involved in the INT reduction method with several marine plankton microorganisms in order to better understand the reduction of the INT by the ETS and other intracellular compounds. We may be able to answer this question in the future.

Reviewer #2: In this paper the authors report on the use of the tetrazolium salt INT to determine respiration rates in microalgal cultures. The use of INT is well established as a mechanism to estimate electron transport activity but the present paper has the novelty of careful attention to the effects of time of incubation and potential toxicity on apparent rates of respiratory activity. As such it does represent a useful contribution to the literature. However there are a number of issues that require attention before the manuscript is suitable for publication.

line 29: I would hardly call use of INT 'recent' as its use as a measure of respiratory electron transport goes back a long way (Ted Packard proposed this in his 1971 paper as did Kenner, R.A. & Ahmed, S.I. Marine Biology (1975) 33: 119 for microalgae and Owens, T.G. & King, F.D. Mar. Biol. (1975) 30: 27. for zooplankton).

This sentence has been changed and now it says: “Several studies have proposed…” (line 28). 

line 58: I am not convinced about the arguments that different cell wall structures represent a major barrier to the entry of INT. For metals at least the cell wall components act less as a permeability barrier than a binding site for ions . The plasmalemma would be more of a barrier to transport of a changed molecule such as INT into the cell. 

There is evidence that the cell wall can be a physical barrier to different molecules (ref 17). In addition, there are studies that reported a lack, or low, INT reduction by specific organisms and suggested differences in the cell wall as one of the possible causes (Thom et al. 1993 J. Applied Bacteriol 74:433-443). We considered that if true, this could create a bias when applying the method to a natural mixed plankton population. We agree with the reviewer that the plasmalemma could also be another barrier. Nevertheless, the significant similarities between the CRO2 – INTR regression lines for each culture suggest that the INT penetrates and it is reduced at a similar rate in the organisms tested.

line 79: Here and elsewhere 'media' is the plural of medium, so only use this (with 'were' ) if you used more than one type of medium. 

The text has been corrected accordingly (line 81). 

line 85: At what stage of growth were these cultures at the time of the experiments? This is important if experiments were done over 2 days as physiological characteristics can change rapidly in batch cultures. There is a brief reference to exponential stage cultures in the Discussion but this needs to be better defined early on. 

As indicated in line 100, the experiments were done in exponential growth phase during both days (day 1: maximum incubation time experiments and day 2: oxygen consumption-INT reduction). 

lines 89/90: which type of PAM? Walz make lots of different forms of the instrument. More details such as saturating pulse intensity are needed 

We have added more information about the Phyto-PAM equipment. Now it reads: 

“The Fv/Fm ratio of the cultures was measured with a Walz PHYTO-PAM Phytoplankton Analyzer fluorometer (Walz, Effelrich, Germany) after cultures were kept for 10 minutes in the dark. The pulse amplitude modulated fluorescence measurements were completed at high repetition rates at four excitation wavelengths (470, 520, 645 and 665 nm). The intensity of the measuring light was 64 μmol quanta m-2 s-1, and the saturating pulse was supplied at the maximum level specified by the Phyto-PAM (intensity level 10) at a pulse of 500 ms.” (Lines 93-99) 

line 102-105: this looks like pseudoreplication rather than the use of independent biological replicates. Please clarify.

The maximum incubation time experiment was done with one cubitainer so there were no biological replicates. Experimental triplicate bottles were taken for each incubation time. We acknowledge that biological replicates would be ideal but the complexity of our experiments precludes this. Each maximum incubation time experiment, as designed here or in other studies (García-Martín et al. 2019, Martínez-García et al. 2009), implies around 14 hours of work. Our main concern was to perform the maximum incubation time experiment as close to the oxygen - INT reduction experiment as possible. To achieve this, we could not perform biological replicates.

lines 122-123: Consider showing the equation for the 3-parameter ‘exponential increase to a limit’ function

The equation has been added (line 132).

lines 208-9: This is fair enough as an estimate of steady state dark respiration, but how does this relate to the fact that respiration rates immediately after a light exposure can be up to x10 those at steady state in the dark? See e.g. Beardall et al a994 Journal of Plankton Research Vol.16 no.10 pp. 1401 -1410. This is also relevant to the Discussion. 

The reviewer is correct when he/she says that there is evidence of an enhancement of respiration following a period of light illumination and that this could bias the respiration rates measured in natural populations over 24 hours, or the respiration measurements from cultures after the light incubation period. Our measurements of respiration are dark respiration: the subsampling was done at the end of the dark cycle and the cells were maintained in dark during all the different processes (similar to a pre-dawn sampling with natural populations). From our results, we have no information regarding the possible stimulation of respiration after the exposure of cells to light. Based on this, we have not included any speculation in the discussion section to as not to distract from the main conclusions: INT did not negatively affect the oxygen consumption when incubated at short incubation times, and there was a similar oxygen consumption – INT reduction relationship for eukaryotic plankton with different cell walls. 

line 366: Stating this is 'for the first time' is not strictly accurate given the mention to reference 12 cited below (line 374). 

Our work is the first to show with different marine eukaryotes that there was no difference between the oxygen consumption of a culture with and without INT during the time when the production of formazan remained linear. Villegas-Mendoza et al. 2015 showed no toxicity (change in the oxygen consumption rate) with only one eukaryotic organism in one experiment, but there was no concurrent estimation of the formazan production. 

line 386: choice of incubation time is further complicated by the enhanced post illumination respiration shown by many phytoplankton species (see comment above). 

We agree with the reviewer. Our results indicate that the choice of the incubation time for the INT reduction methodology depends not on the cell biomass, but on the activity of the sample, and that it is important to calculate the maximum INT incubation time to prevent biased the measurements. 

Reviewer #3: The manuscript "INT reduction is a valid proxy for eukaryotic plankton respiration despite the inherent toxicity of INT and differences in cell wall structure" focuses on an alternate method for measuring plankton respiration with the help of INT. Overall the manuscript is well written and has applicable objective, however the authors would need to address the following concerns:

1. Tetrazolium salts can also be reduced by reactive oxygen species in the cell, therefore if the plankton is under oxidative stress, use of this technique can lead to over-estimation of the respiratory rates. The authors need to either compare the reduction of INT in the presence and absence of oxidative stress or at least discuss the possibility of interference by ROS and advise caution with the interpretation.

We are afraid we cannot quantify the proportion of INT reduced by ROS as we did not estimate the reduction of INT in the presence and absence of oxidative stress. We have commented on the possibility of biases due to the presence of ROS in the text and have suggest caution in interpreting the results (lines 444-457). 

2. Fig. 3 shows extreme variation in results even within the same species, for example the oxygen consumption patterns are very different between the two figures of T. pseudonana. This suggest extreme variability in this method, not just between species but within the same species. Can the authors provide an explanation for this?

Also, it looks like E. huxleyi and Scrippsella sp. were excluded from this measurement, is there a particular reason?

We agree with the reviewer that there are different patterns for the oxygen consumption between species and within the same species, for example T. pseudonana. The two examples in Figure 3 are from different experiments with different cell abundances and different respiration rates. Different trends in oxygen consumption estimated with oxygen sensors have been previously reported (Wikner et al. 2013 L&O: methods 11: 1-15, Briand et al. 2004 L&O: methods 2: 406-416) which indicate that dark respiration does not always follow a linear decrease. However, there was a linear decrease during the incubation time employed for the estimation of the oxygen consumption by Winkler, and therefore, our oxygen consumption results should not be biased by a non-linear trend. This is indicated in lines 241-246. We are afraid we cannot provide any oxygen consumption data measured with the optodes during E. huxleyi experiments. There was a reaction between the metallic needle of the optode and the medium, which precluded accurate measurements. The optode sensors were returned to the manufacturer for servicing and re-coating with a non-reactive film. The optodes were not available for the INT maximum incubation time experiments with Scrippsiella sp. 

3. Measurement of respiration with the help of Clark-type oxygen electrode usually takes about 10-15 mins? How does the authors just the use of INT as a better way to measure respiration? Moreover, the authors themselves have used optodes to compare the INT measurements. What advantages does INT measurements have over clark type electrodes and optodes? Consider discussing this in terms of cost, time, and sample volume?

The reviewer is correct when he/she comments that Clarke type electrodes and oxygen optodes are sensors that can provide oxygen concentration measurements in short periods of times (0.5-2 hours). However, despite their fast response, they still have limitations regarding the response time needed when applied to low plankton respiration rates (~ 0.5 μmol L-1 d-1 as occurs in open ocean oligotrophic environments). In these situations, the time needed to record the change is longer, and always longer than the time needed for the incubation in the INT reduction technique. Advantages of the INT technique (as mentioned in lines 31-35) over optodes and electrodes include sensitivity and the ability to determine the respiration of size classes of the plankton community without pre-incubation filtration.

We do not agree that a detailed comparison of the different methods used to estimate plankton respiration (oxygen consumption of an incubated water sample measured by Winkler and oxygen sensors; in vitro ETS activity and INT reduction and 14CO2 production during incubations with 14C labelled compounds) in terms of cost, time and sample volume is appropriate in this manuscript. We would prefer to focus on the two conclusions: INT did not negatively affect the oxygen consumption when incubated at short incubation times, and there was a similar oxygen consumption – INT reduction relationship for eukaryotic plankton with different cell walls.

---

## [Editor Report · Decision Letter 1]

18 Nov 2019

INT reduction is a valid proxy for eukaryotic plankton respiration despite the inherent toxicity of INT and differences in cell wall structure

PONE-D-19-27198R1

Dear Dr. Garcia-Martin,

We are pleased to inform you that your manuscript has been judged scientifically suitable for publication and will be formally accepted for publication once it complies with all outstanding technical requirements.

With kind regards,

Antonietta Quigg, PhD

Academic Editor

PLOS ONE

Additional Editor Comments (optional):

In submitting the final manuscript, please consider inserting, as appropriate, additional parts of your responses to the reviewers concerns. for example it will be helpful for readers to know why you had no biological controls, the issues with the optodes and why some data is missing, etc... in each case... a carefully placed sentence in the methods and results could further enhance the paper.

---

## [Editor Report · Acceptance letter]

2 Dec 2019

PONE-D-19-27198R1 

INT reduction is a valid proxy for eukaryotic plankton respiration despite the inherent toxicity of INT and differences in cell wall structure 

Dear Dr. Garcia-Martin:

I am pleased to inform you that your manuscript has been deemed suitable for publication in PLOS ONE. Congratulations! Your manuscript is now with our production department. 

With kind regards,

on behalf of

Dr. Antonietta Quigg 

Academic Editor

PLOS ONE